# Effects of Meteo-Climatic Factors on Hospital Admissions for Cardiovascular Diseases in the City of Bari, Southern Italy

**DOI:** 10.3390/healthcare11050690

**Published:** 2023-02-26

**Authors:** Vito Telesca, Gianfranco Castronuovo, Gianfranco Favia, Cristina Marranchelli, Vito Alberto Pizzulli, Maria Ragosta

**Affiliations:** 1School of Engineering, University of Basilicata, Viale dell’Ateneo Lucano 10, 85100 Potenza, Italy; 2Interdisciplinary of Medicine, School of Medicine, University of Bari, Piazza Giulio Cesare 11, 70124 Bari, Italy; 3Freelance Engineer, Via Mazzini 54, 75025 Policoro, Italy

**Keywords:** hospital admission, cardiovascular diseases, temperature, distributed lag non-linear model, time series decomposition, feature importance, random forest

## Abstract

The objective of this study was to determine the relationship between weather conditions and hospital admissions for cardiovascular diseases (CVD). The analysed data of CVD hospital admissions were part of the database of the Policlinico Giovanni XXIII of Bari (southern Italy) within a reference period of 4 years (2013–2016). CVD hospital admissions have been aggregated with daily meteorological recordings for the reference time interval. The decomposition of the time series allowed us to filter trend components; consequently, the non-linear exposure–response relationship between hospitalizations and meteo-climatic parameters was modelled with the application of a Distributed Lag Non-linear model (DLNM) without smoothing functions. The relevance of each meteorological variable in the simulation process was determined by means of machine learning feature importance technique. The study employed a Random Forest algorithm to identify the most representative features and their respective importance in predicting the phenomenon. As a result of the process, the mean temperature, maximum temperature, apparent temperature, and relative humidity have been determined to be the most suitable meteorological variables as the best variables for the process simulation. The study examined daily admissions to emergency rooms for cardiovascular diseases. Using a predictive analysis of the time series, an increase in the relative risk associated with colder temperatures was found between 8.3 °C and 10.3 °C. This increase occurred instantly and significantly 0–1 days after the event. The increase in hospitalizations for CVD has been shown to be correlated to high temperatures above 28.6 °C for lag day 5.

## 1. Introduction

Climate changes and climate seasonal variability affect human health. Meteorological factors, such as temperature, relative humidity, and atmospheric pressure, determine several negative health outcomes. Exploring relations between health and weather conditions at a local scale may allow us to measure climate change impacts on the population. Several studies show the correlation existing between ambient temperature and mortality or morbidity [1,2]. Since the last century, high temperatures and heat waves have been associated with excess deaths in many US cities [3] and were recognized as important factors determining deaths, chronic bronchitis, pneumonia, ischemic heart disease, and cerebrovascular disease in England and Wales [4]. Increased mortality associated with high average temperatures was found in Seoul, Beijing, Tokyo, and Taipei in Asia [5]. Temperature and mortality have a complex relationship, influenced by geographic, climatic, and demographic factors [6]. The vulnerability of populations to temperatures can be influenced by social, economic, demographic, and infrastructural variables and, for this reason, developing countries are more sensitive to climate change [7]. For the African continent, a significant correlation between temperature increase and an increase in mortality and morbidity for cardiovascular diseases was shown [7,8,9,10,11,12]. Even on the European continent, the implications of climate change on human healthcare were addressed [13,14,15,16,17,18]. The project “Assessment and prevention of acute health effects of weather conditions in Europe” (PHEWE project) was an attempt to examine the influence of temperature on various mortality and morbidity outcomes utilising a standardised approach [19]. The project analysed acute health impacts of highly variable climatic conditions, both during hot and cold seasons in numerous European countries. It was shown that, in the short term, temperature and relative humidity were strongly correlated with hospital admissions and mortality [20]. Although the correlation between high temperatures and mortality is clear, there is less evidence of the impact of high temperatures on hospitalisation around the world [21]. Several studies have shown that high temperatures are associated with increased hospitalisation rates for both cardiovascular and respiratory diseases in several cities in the United States of America [22]. The short-term effect of temperature on respiratory diseases was evident also for children [23]. The potential influence of the environment on the infarct is underlined, analogous to considerations regarding the increase in stroke risk. At lowering temperatures, the percentage of attacks would increase by 195% in winter and 10% in spring. In this case the cold would favour the formation of blood clots with a consequent increase in risk in patients suffering from fibrillation. Investigation into the relationships between environment and pathologies could help in implementing preventive measures such as anticoagulant therapies and a reduction of exposure to cold. The possibility of predicting events linked to cardiovascular diseases, combined with greater attention to lifestyles and the living environment, suggests a benefit to deeply investigating the effects of the environment and climate on the risk of cardiovascular diseases through ad hoc therapeutic strategies. Moreover, it is possible to reduce the economic costs related to these events. Rising temperatures and the concentration of pollutants in the atmosphere also have repercussions on respiratory diseases. Climate change acts by leading to an increase in ozone and fine particulate levels, generating an increase in terms of morbidity and mortality. Heat mainly may affect a pool of fragile individuals in which death or the onset of the disease is anticipated by a short period of time [22]. The mortality rate or morbidity is influenced not only by the current day’s temperature but also by the temperature of previous days [24]. Distributed lag models have been applied to explore the delayed effect of temperature on mortality [20,25,26]. To overcome the strong correlation among daily temperatures on short periods, constrained distributed lag structures are used in time series regressions [27]. Estimates are constrained by the use of smoothing methods, such as natural cubic splines or polynomials, but both unbound and constrained distributed lag models presume a linear relationship between temperature and mortality, making them weak for well characterising the influence of temperature on mortality. Distributed lag non-linear model (DLNM) has been developed to simultaneously estimate the non-linear and delayed effects of temperature (or air pollution) on mortality (or morbidity). Using this model, a three-dimensional plot allows us to show the relative risks both for temperature and for delays [27,28]. Cardiovascular disease (CVD) is the leading cause of mortality, morbidity, and disability in Europe and specifically in Italy, requiring greater attention to cardiovascular risk factors in health planning and resource allocation [29]. For this reason, we have focused our attention on these specific pathologies in order to identify meteo-climatic parameters strongly correlated to the incidence of daily hospitalizations. In this context we apply a methodological procedure for analysing and modelling the relationship between meteo-climatic factors and the daily hospitalizations for CVD in the city of Bari, southern Italy. The purpose is to identify the meteorological parameters that drive admissions to the emergency room for cardiovascular diseases. Furthermore, the relative risk of the onset of this type of pathology, concerning all the selected meteorological variables, will be analysed. In this way, it will be possible also to improve the management of access flows to the emergency room.

## 2. Materials and Methods

### 2.1. Study Area and Data Collection—Hospitalisation Data and Preliminary Statistical Analysis

Hospitalisation data analysis involves daily accesses to the emergency room in the Bari Policlinico “Giovanni XXIII” for the 2013–2016 reference four-year period. Bari is the ninth Italian municipality in terms of population size, the third most populous municipality in southern Italy after Naples and Palermo with almost 300,000 inhabitants. The database of daily admissions in the emergency room was formatted according to the compilation scheme: first name; surname; sex; date of birth; birthplace; place of residence; citizenship; day, month, year of acceptance; acceptance time; main problem; day, month, year of discharge; discharge procedure; observation methods; hospitalisation department. Particularly, pathology and/or symptomatology was classified on the basis of 33 codes (Table 1).

In Table 2 and in Figure 1, for each year, the number of admissions, divided for genders, is summarised. In particular, for the year 2013, 75,927 entries into the emergency room were counted, of which, 40,265 were male patients, 35,032 were female patients, and 630 data cases were missing inherent in sex. For the year 2014, a total of 80,690 admissions were counted, of which, there were 42,554 referring to the male gender, 37,127 female, and 1009 cases of missing data. For the year 2015, the total number of admissions to the emergency room amounts to 75,334, of which, 40,091 were men, 34,327 were women, and 916 were cases of missing data. For the year 2016, 71,550 visits to the emergency room were registered, of which, 38,007 were male patients, 32,914 were female patients, and 629 were cases of missing data. For this study, only the data associated with cardiovascular pathologies were selected and examined. According to the ESC (European Society of Cardiology), the high rate of deaths (in Europe there are about 4 million a year) caused by cardiovascular diseases and the correlation of these with decreasing temperatures are strongly evident. The ESC stresses that cardiovascular disorders are the leading cause of death in all European countries, especially for women; moreover, congenital heart disease alone is the leading cause of death below 65 years. Canadian and Taiwanese studies have shown that each 10 °C reduction in atmospheric temperature corresponds to an increase of 7% in the myocardial infarction (sudden rupture of a coronary artery plate) and that a reduction of 5 °C corresponds to an increase equal to 13% in the risk of thrombo-embolic fibrillation stroke. In the case of the heart attack, the study was able to identify the possibility of preceding the event 2 days in advance by observing the temperature trend, which, if they remain below 0 °C in the daytime hours, the risk of heart attack increases.

Therefore, considering cardiovascular diseases, we select only the admissions with the codes shown in Table 3.

For each year, the distribution of cardio-vascular admissions in the emergency room is shown in Table 4 and in Figure 2. In Figure 3 the frequencies of different codes in CVD are shown. Furthermore, for epidemiological purposes, it is important to separate data admission on the basis of age of the patient (Figure 4 and Table 5).

As shown in Figure 4, the 40–54 age group has the highest number of hospital admissions. Typically, the over-60-years and the over-70-years are the categories most affected by cardiovascular diseases, both for the severity of the disease and for the greater risk of complications caused by the presence of other diseases. In this case, instead, the highest incidence is observed in more young people.

### 2.2. Meteo-Climatic Parameters and Preliminary Statistical Analysis

The meteo-climatic data for the investigated period were collected by ARPA of the Apulia region. For the monitoring activities, ARPA manages a Telemetric Network, with five automatic stations. Each automatic meteorological station includes the following: an “ECO2” series acquisition unit with 8 analogue inputs, which controls the system and provides for the acquisition, pre-processing, and storage of data; a software package (Ecodata32) dedicated to the management of the survey stations and able to dialogue with the stations and to manage and process the data; sensors consisting of electronic or mechanical devices that measure a specific meteorological parameter. The data are recorded with a half-hourly frequency and are always expressed in solar time. In this study, the meteo-climatic parameters taken into account are as follows: average daily minimum temperature (Tmin); average maximum daily temperature (Tmax); average temperature (Tmean); temperature at the dew point (Tdewp); apparent or perceived temperature (Tapp); atmospheric pressure (P_atm); average relative humidity (RH); and average absolute humidity (AH). In Table 6, the descriptive statistics for each variable are summarised.

### 2.3. Methodology

In order to determine the meteorological variables that most influence admissions to the emergency room for cardiovascular disease and to calculate the related relative risk, the methodology illustrated in Figure 5 was adopted. It is composed of two phases: Phase 1: correlation analysis and Phase 2: application of Machine Learning feature importance and DLNM model.

In Phase 1, the pair dependence between meteorological variables and admissions to the emergency room for cardiovascular diseases is analysed with means of correlation analysis (see Section 3.1).

If the Pearson coefficient *r* is higher or equal than a prefixed threshold (0.45 in this case) and the *p*-value is lower than 0.01, Phase 2 will be carried out.

Conversely, the trend components are extracted using the Seasonal and Trend decomposition via Loess (STL) (see Section 3.2), and Phase 2 will be carried out using trend components data.

In Phase 2, a feature importance procedure is applied (with artificial intelligence techniques, see Section 3.3) to determine the most significant meteorological variables, and then the DLNM model [28] is applied to estimate the related relative risk (see Section 3.4).

## 3. Results

### 3.1. Correlation Analysis

Correlation analysis shows how the features are related to each other or with the target variable. Positive correlation indicates that an increase in one feature’s value increases the value of the target variable, whereas negative correlation means that an increase in one feature’s value reduces the value of the target variable. A correlation matrix was used calculating the Pearson Correlation Coefficient *r*, for each pair of quantitative features. [30] The Pearson’s correlation between any two variables *x*,*y* is:$$r_{xy}=\frac{\sum_{i=1}^{n}\left(x_{i}-\underline{x}\right)\left(y_{i}-\underline{y}\right)}{\sqrt{\sum_{i=1}^{n}{\left(x_{i}-\underline{x}\right)}^{2}}\sqrt{\sum_{i=1}^{n}{\left(y_{i}-\underline{y}\right)}^{2}}}$$
where:*n* is the sample size;*x_i_, y_i_* are the individual sample points and $\underline{x}$, $\underline{y}$ are the sample means.

Each cell of the matrix receives a single number from −1 to +1; therefore, the table shows the strength of the (linear) relationship between any two features. The correlation analysis is also based on the *p*-value. In null hypothesis significance testing, the *p*-value is the probability of obtaining test results at least as extreme as the results actually observed, under the assumption that the null hypothesis is correct. A very small *p*-value means that such an extreme observed outcome would be very unlikely under the null hypothesis. This statistical measure defines the reliability of the values obtained from the correlation, as it helps to understand if the results of an experiment fall within the normal range of values for the event under observation. Only the *p*-value values less than 0,01 and *r* greater than or equal to 0.45 were considered for the hypothesis of acceptability for the set of input features. In Table 7, the correlation matrix among meteorological parameters and CVD admissions is shown. We note that (last column of the Table 7), for CVD, all the *r*-values satisfy the *p*-test.

Moreover, we note that for all the pairs, CVD admission and meteorological parameters *r*-values are lower than 0.45, the fixed threshold.

### 3.2. Decomposition Model

The Seasonal and Trend decomposition using Loess (STL) is a filtering procedure for decomposing a time series into seasonal, trend, and remainder components [31]. The trend component is the low frequency variation in the data together with the nonstationary, long-term changes level. The seasonal component is the variation in the data at or near the seasonal frequency. The remainder component is the remaining variation in the data beyond that in the seasonal and trend components. Suppose the data, the trend component, the seasonal component, and the remainder component are denoted by *Y_v_*, *T_v_*, *S_v_*, and *R_v_*, respectively, for *v* = 1 to N; then:$$Y_{v}=T_{v}+S_{v}+R_{v}$$

The STL model uses robust local-weighted regression as a smoothing method for time series decomposition. When estimating the value of a response variable, a subset of data is selected from the vicinity of the predicted variable, and then linear or quadratic regression is performed on the subset by using the weighted least squares method to reduce the weight of the value far from the estimated point. Finally, the value of the response variable can be estimated by the local regression model. This point-by-point method is generally used to fit the whole curve to decompose the time series accurately. The aim was to identify a simple method to verify that signal decomposition would lead to a noticeable improvement in results and allow the use of ML techniques for simulating these processes. This was accomplished by verifying the validity of STL and evaluating the stationarity of the data series using the Dickey–Fuller method.

In Table 8, we compare the absolute values of *r* calculated among CVD admissions and meteorological parameters before and after the application of the decomposition model. As you can note, the variables Tmean, Tmax, Tapp, and RH show *r*-values greater than the threshold value 0.45, and only for atmospheric pressure (P_atm) is the *p*-value test not verified; therefore, it will be removed from the following analysis.

### 3.3. Application of Feature Importance

Building a ranking of features is useful for better understanding the data and better understanding a model. Given an external estimator that assigns weights to features, Feature Importance calculates relative importance of the variables, enabling the identification of the features that have the most impact on the simulation of the phenomenon. A Random Forest was used here as an external estimator to determine the relative importance of all features. The aim is to quantify the strength of the relationship between the predictors and the outcome. The higher the score, the more important or relevant is the feature towards the output variable.

The Figure 6 shows the Relative Importance (RI) of the meteo-climatic features obtained with a Random Forest model in our case. The graph shows that the variable Tmax alone explains over 40 percent of the model’s values, RH is over 20%, Tapp is about 20%, and Tmean is about 10%.

### 3.4. Application of Distributed Lag Non-Linear Model (DLNM)

DLNMs are statistical methods developed for time series data and used to describe the additional time dimension of the exposure–response relationship determining the distribution of next effects after the occurrence of events (in lag times). Several studies have shown how DLNM simultaneously estimates the nonlinear and delayed effects of temperature on mortality or morbidity [27,28]. This statistical framework rests on the definition of a “cross-base” function, a two-dimensional functional space expressed by the combination of two sets of basic functions, which specify the relationships in the dimensions of predictor and delays [28]. In order to model the shape of the non-linear relationship in each of the two spaces we are considering, that of the predictor and the lags, we must simultaneously apply two transformations:Choose a basis for x (vector of the exposures) such as to define the dependence in the space of the predictor, specifying the basis matrix Z obtained by applying the basis functions to x;Create the additional delay dimension for each of the derived base variables of x stored in Z.

This operation produces an array representing the lagged occurrences of each base variable x. Despite its complicated parameterization, estimating and inferring the parameters of a DLNM is no more difficult compared to any other generalised linear model and can be performed using standard statistical software after cross-base variables have been provided [28]. This application preserves the hypothesis of non-linearity of the exposure–response relationship and the hypothesis that the exposure is variable over time while maintaining the algebra of the DLNM unchanged except for the use of smoothing functions for time series. Decomposition of the time series into trend components allows for modelling the non-linear relationship of exposure–response with the DLNM. A generalised linear regression model with quasi-Poisson distribution, combined with the DLNM, was used to fit the relationship between the trend components of daily CVD hospital admissions and meteo-climatic factors. The fits of models with different response variable specifications were compared using AIC and BIC, identifying the most suitable model and evaluating the evidence for a non-linear exposure response as well as the consistent risk along the lag. The results obtained show that the response variables best correlated to the phenomenology are the trend components of Tmax, Tapp, and RH.

The DLNM model becomes the following:$$Y=Poisson\;\left(\mu_{t}\right)\;Log\left(\mu_{t}\right)=\alpha +\beta Tmax_{t,l\;}+\;Tapp+RH+time+Dow$$
where *µ_t_* is the trend component of daily CVD hospital admissions at calendar day *t* (*t* = 1, 2, 3, …, 1447); *α* is the intercept; *Tmax_t,l_* is the cross-basis matrix produced by DLNM [28] (Gasparrini et al., 2010). This matrix is obtained by the combination of the exposure–response function with three internal knots placed at the 10th, 75th, and 90th percentiles of the maximum temperature distributions and the lag–response function modelled with three internal knots placed at equally spaced values in the log scale. According to previous studies, the maximum lag was set up to 21 days for effects of cold temperature which appeared only after some delay and lasted for several days; the trend components of relative humidity (*RH*) and apparent temperature (*Tapp*) were used as response variables; Day of the week *(Dow*) was also included in the model as indicator variables [32,33]. The median value of temperature (20 °C) was defined as the baseline temperature (centring value) for calculating the RR [34,35,36]. All the analyses were performed with the software R, version 4.0.4, using the “dlnm package”, available on the R comprehensive archive network (CRAN). The package contains functions for building basic matrices for specifying DLNM and then for predicting and tracking results for a fitted model. The expected effect was explained according to the Relative Risk parameter (RR). RR represents the probability that a subject, belonging to a group exposed to certain factors, develops the disease, with respect to the probability that a subject belonging to an unexposed group develops the same disease. This index is used in cohort studies where exposure is measured over time. If the RR is equal to 1, the risk factor is irrelevant to the appearance of the disease; if the RR is greater than 1, the risk factor is implicated in the onset of the disease; if the RR is less than 1, the risk factor defends against the disease (defence factor). The results were expressed in terms of percentage increase and respective 95% confidence intervals.

The two-dimensional relationship of exposure–response estimated with DLNM can be graphically summarised in 3D (Figure 7) and contour plot (Figure 8). The distributed nonlinear lag surface revealed a non-linear relationship between temperature and hospital admissions for cardiovascular diseases. In general, the lag patterns for hot and cold effects showed statistically positive but not significant cold effects occurred, while hot effects were strong and not correlated significantly. The cold effects followed a pattern of increasing RR on the current day or on lag day 0–1.

In order to provide a specific assessment of the dose–response curve, the cumulative effects of temperatures at lag 0, 5, 15, and 20 days and by lag at specific temperatures 8.3 °C, 10.3 °C, 10.9 °C, and 30.9 °C corresponding to 0.1th, 5th, 95th, and 99.9th percentiles of temperature distribution are reported in Figure 9.

## 4. Conclusions and Remarks

This study examines the correlations between the meteo-climatic factors and hospitalizations for cardiovascular diseases in the city of Bari. Correlations previously identified in epidemiological studies regarding the exposure–response relationship between daily visits to the emergency room and the variation of some meteo-climatic parameters suggest that morbidity in the case of cardiovascular diseases was related to the lowering of the average seasonal temperature. Our results confirm this relationship by evaluating a forecast scenario that shows an increase in the relative risk of hospitalizations as a function of the delayed effects of time. The correlation analysis carried out after the time series data decomposition highlights that the number of daily admissions to the emergency room for cardiovascular diseases and the daily parameters of maximum temperature, apparent temperature, and relative humidity are strongly related. A machine learning methodology, including Feature Importance, has mathematically validated the selection of the more relevant meteo-climatic variables to be used in the statistical model for the definition of a possible risk scenario, reducing the overfitting and, consequently, reducing the variance of the data. The non-linearity of the exposure–response relationship was, therefore, addressed through the application of the DLNM, using the trend components as input data. The prediction analysis carried out on decomposed time series shows an evident but not significant increase in the relative risk associated with colder temperatures between 8.3 °C and 10.3 °C. The effect, as highlighted in Figure 7, Figure 8 and Figure 9, occurs instantly and significantly between 0–1 days after the event and subsequently 4–10 days later. In agreement with other studies, the sample size may have contributed to underestimating the cold effect in the percentage increase in relative risk. The effect of temperature on cardiovascular diseases has been shown to be evident with the lowering of the seasonal temperature averages; however, most of these studies were conducted on at least a ten-year timeframe of observations. The increase in hospitalizations for CVD has been shown to be correlated to temperatures above 28.6 °C for lag day 5. This correlation does not appear to be significant; the increase in percentage terms of the relative risk associated with high temperatures is 0.73 (95% CIs), and this value is very low compared to the forecasted statistical scenario. Previous results suggest that the scenario produced by the application of the DLNM has identified an increased relative risk of hospitalisation due to lower temperatures. The sensitivity of the phenomenon to hot days has been shown to be very low. The study highlighted the delayed effects in terms of lag days. The results obtained do not show very high-risk percentages due to the numerosity of the input data. The small number of observations may have contributed to underestimating the risk percentages obtained. This study provides encouraging results that validate the extent of the influence of weather–climatic parameters on human health. Although the forecast scenario shows a lower percentage increase in RR compared to the reference scenarios of other studies, our methodological procedure for selecting the predictors and the evaluation of their influence on daily access to the emergency room appears to be reliable and in line with epidemiological studies. Furthermore, the results confirm the applicability of the DLNM method. Our findings align with those of many other studies [37] who utilised the DLNM to study the relationship between PM2.5 exposure, temperature, and health outcomes in five cities in Poland. Their findings showed the effectiveness of the DLNM approach, with PM2.5 being identified as the most significant pollutant. Additionally, several other studies [38,39,40,41] in China have leveraged the DLNM method to investigate the connection between PM2.5 exposure, temperature, and human health outcomes, further demonstrating the capabilities of the DLNM approach in these types of analyses. Furthermore, [42] confirmed that this procedure is able to characterise the complex pattern existing among environmental variables and human health.

In conclusion, our findings confirm that there is a noticeable correlation between the variation in meteorological parameters and the daily hospitalizations for cardiovascular diseases. Longer time series would allow further confirmation of the results and to identify specific variation ranges of other meteorological parameters, not only of temperature in which the relative risk increases. Moreover, more advanced signal decomposition techniques, such as the Ensemble Empirical Mode Decomposition (EEMD) [43,44,45], will be applied, for improving our analysis and for uncovering further insights into the relationship between the environmental features and cardiovascular diseases or other negative impacts on human health.

## Figures and Tables

**Figure 1 healthcare-11-00690-f001:**
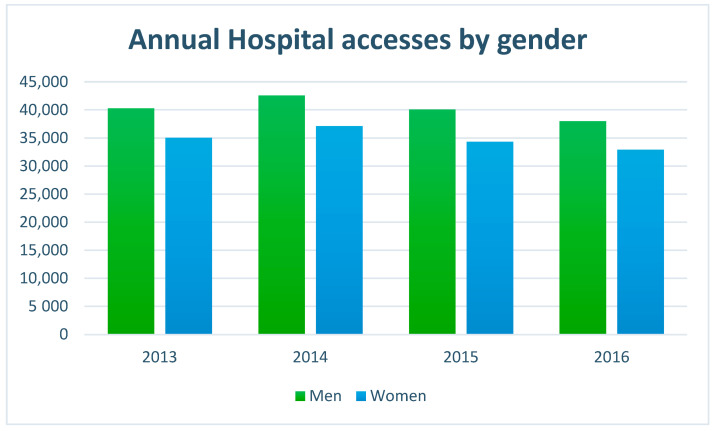
Annual hospital admissions in ER by gender.

**Figure 2 healthcare-11-00690-f002:**
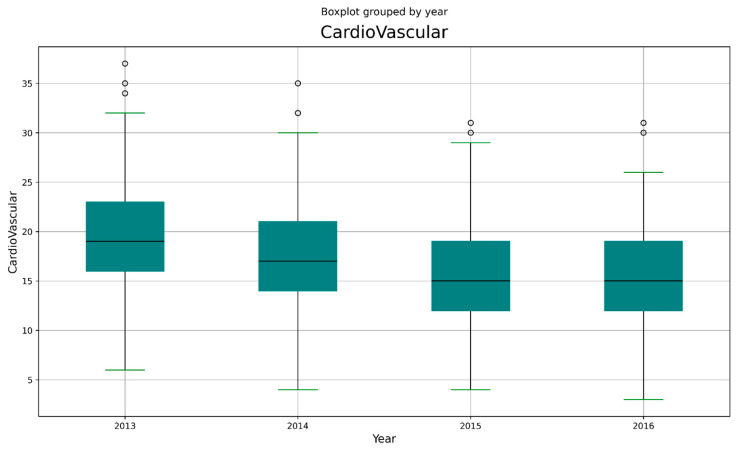
Boxplot of CVD admissions.

**Figure 3 healthcare-11-00690-f003:**
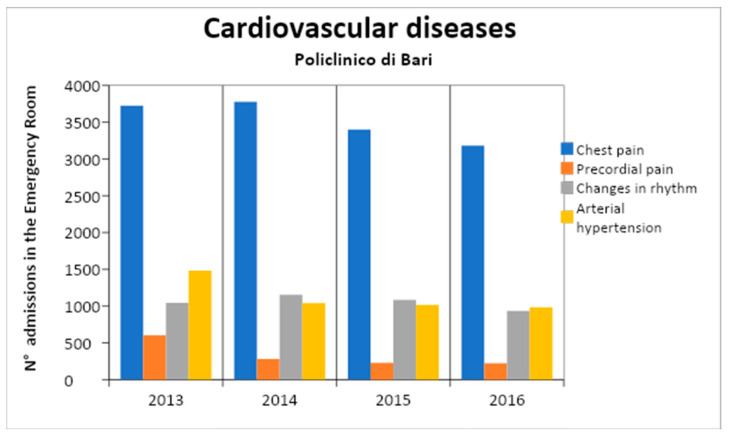
Annual admissions for CVD.

**Figure 4 healthcare-11-00690-f004:**
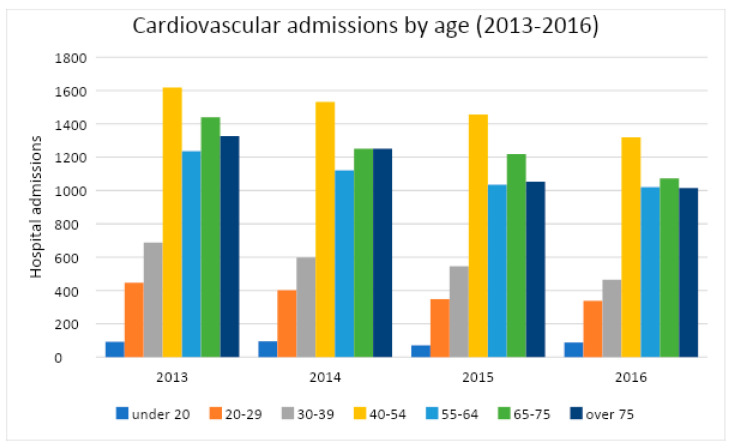
CVD admissions classified by age class.

**Figure 5 healthcare-11-00690-f005:**
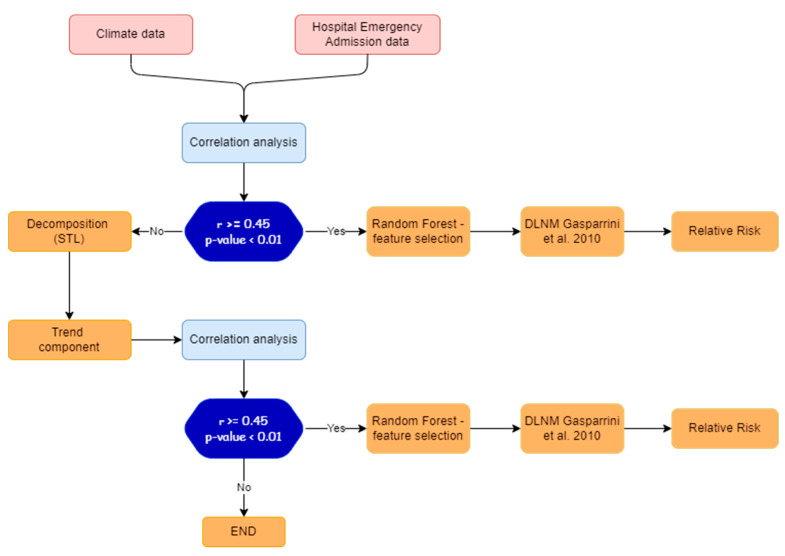
Experimental design of methodological procedure.

**Figure 6 healthcare-11-00690-f006:**
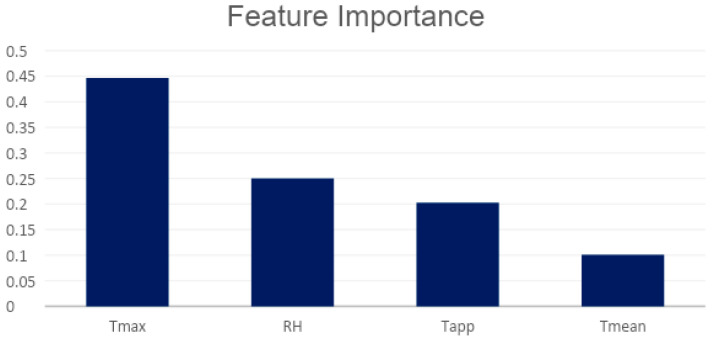
Relative importance of selected features by correlation analysis.

**Figure 7 healthcare-11-00690-f007:**
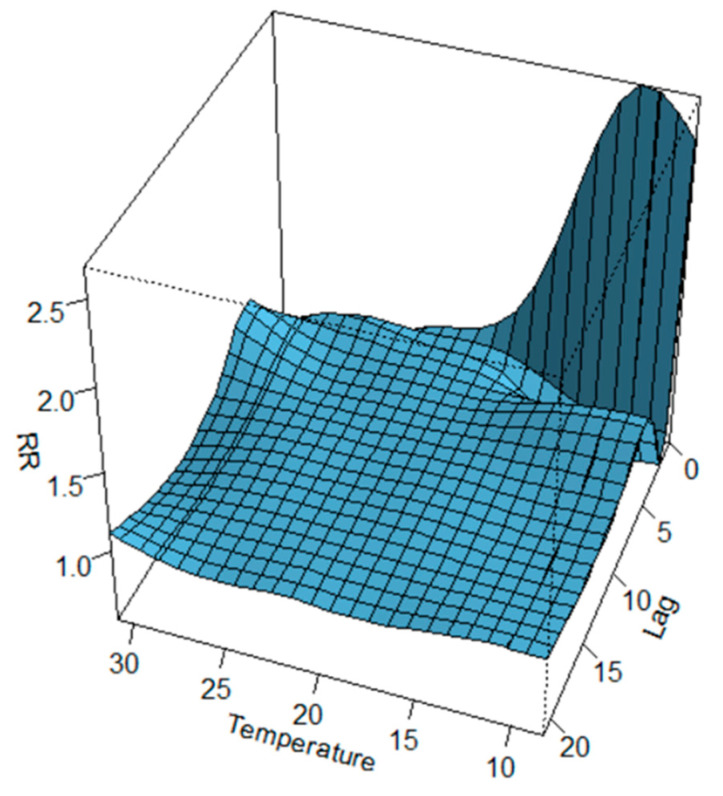
3D plot of RR of CVD hospital admissions along temperature and lags with references of 20 °C.

**Figure 8 healthcare-11-00690-f008:**
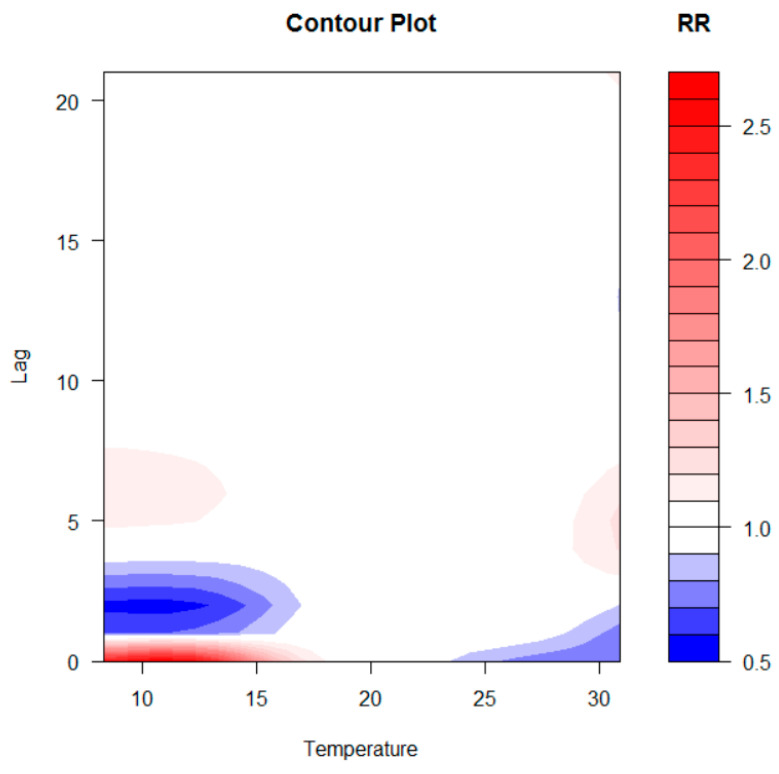
Contour plot of RR of CVD hospital admissions along temperature and lags with references of 20 °C.

**Figure 9 healthcare-11-00690-f009:**
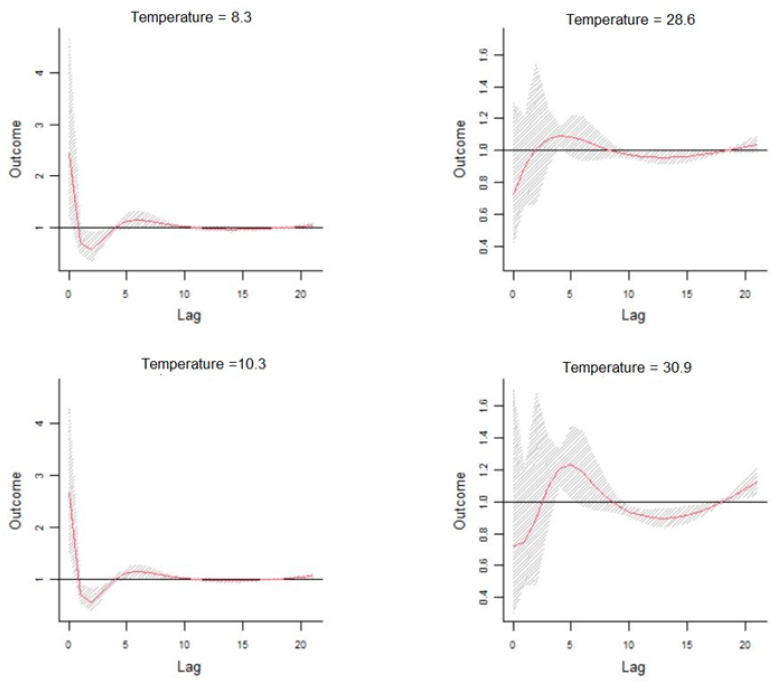
Dose–response curve cumulative effects of temperatures at different time-lags.

**Table 1 healthcare-11-00690-t001:** Main symptoms entering the emergency room and identification code.

CODE	Main Problem/Symptomatology	CODE	Main Problem/Symptomatology
1	Coma	18	Oto rhino laryngeal symptoms or disorders
2	Acute neurological syndrome	19	Obstetric-gynaecological symptoms or disorders
3	Other nervous system symptoms	20	Dermatological symptoms or disorders
4	Abdominal pain	21	Odontostomatological symptoms or disorders
5	Chest pain	22	Urological symptoms or disorders
6	Dyspnea	23	Other symptoms or disorders
7	Precordial pain	24	Legal-medical investigations
8	Shock	25	Social problem
9	Non-traumatic haemorrhage	26	Fall from high
10	Trauma	27	Scalding
11	Intoxication	28	Psychiatric
12	Fever	29	Pneumology-Respiratory pathology
13	Allergic reaction	30	Violence from other
14	Changes in Rhythm	31	Self-harm
15	Hypertension	98	Dehydration
16	Psychomotor agitation	99	Animal bite
17	Eye symptoms or disorders		

**Table 2 healthcare-11-00690-t002:** Number of admissions in the Emergency Room (ER) divided for gender.

Gender	2013	2014	2015	2016	Total
Men	40,265	42,554	40,091	38,007	160,917
Women	35,032	37,127	34,327	32,914	139,400
No data	630	1,009	916	629	3184
Total	75,927	80,690	75,334	71,550	303,501

**Table 3 healthcare-11-00690-t003:** Selected codes for cardiovascular diseases.

Code	Specific Problem	Classification
5	Chest pain	Cardiovascular diseases
7	Precordial pain	
14	Changes in Rhythm	
15	Hypertension	

**Table 4 healthcare-11-00690-t004:** Number of admissions in ER for CVD diseases.

Cardiovascular.	2013	2014	2015	2016
No. of admissions	6854	6252	5728	5319
CVD admissions (%)	9.0	7.7	7.6	7.4

**Table 5 healthcare-11-00690-t005:** Admissions for CVD classified by age class.

Age Class	2013	2014	2015	2016
under 20	92	95	71	88
20–29	447	401	348	338
30–39	688	597	545	464
40–54	1617	1532	1456	1320
55–64	1236	1122	1035	1020
65–75	1440	1251	1218	1073
over 75	1326	1250	1053	1016
No Data	8	4	2	0
Total	6854	6252	5728	5319

**Table 6 healthcare-11-00690-t006:** Descriptive data statistics. Legend: avg = average; std = standard deviation; 25%, 50%, and 75% = 25th, 50th, and 75th percentiles, respectively; min–max = range.

	Tmin	Tmean	Tmax	Tdewp	Tapp	P_atm	RH	AH
	(°C)	(°C)	(°C)	(°C)	(°C)	(mbar)	(%)	(%)
avg	16.1	17.7	19.1	12.4	23.3	1008.2	72.2	11.3
std	6.4	6.1	6.3	5.4	8.7	8.6	10.9	3.6
min	0.0	3.5	3.7	−4.2	3.0	976.6	37.0	3.5
25%	11.0	12.2	14.0	8.2	15.8	1003.0	65.0	8.3
50%	16.0	17.3	19.0	12.5	22.2	1007.3	73.0	10.8
75%	20.8	22.8	24.1	16.9	30.2	1014.0	80.0	14.0
max	30.8	32.0	37.0	26.0	52.8	1042.0	99.0	24.1

**Table 7 healthcare-11-00690-t007:** Correlation matrix; the asterisks indicate the cases in which the hypothesis null on *p*-value is not satisfied.

*r*	Tmean	Tdewp	Tapp	Tmin	Tmax	P_atm	RH	AH	CVD
Tmean	1	0.91	0.99	0.94	0.95	−0.13	−0.38	0.90	−0.25
Tdewp	0.91	1	0.91	0.88	0.84	−0.14	0.03 *	0.99	−0.21
Tapp	0.99	0.91	1	0.91	0.95	−0.12 *	−0.35	0.90	−0.25
Tmin	0.94	0.88	0.91	1	0.80	−0.27	−0.30	0.87	−0.18
Tmax	0.95	0.84	0.95	0.80	1	0.02 *	−0.40	0.82	−0.28
P_atm	−0.13	−0.14	−0.12 *	−0.27	0.02 *	1	0.01 *	−0.14	−0.14
RH	−0.38	0.03 *	−0.35	−0.30	−0.40	0.01 *	1	0.03 *	0.15
AH	0.90	0.99	0.90	0.87	0.82	−0.14	0.03 *	1	−0.22
CVD	−0.25	−0.21	−0.25	−0.18	−0.28	−0.14	0.15	−0.22	1

**Table 8 healthcare-11-00690-t008:** Correlation between meteorological parameters and CVD admissions, with absolute value of Pearson’s coefficient (|r|) before (CVD_1_) and after (CVD_2_) decomposition model application.

Variable	CVD_1_	CVD_2_
Tmean	0.25	0.47
Tdewp	0.21	0.42
Tapp	0.25	0.47
Tmin	0.18	0.36
Tmax	0.28	0.54
P_atm	0.14	0.30
RH	0.15	0.45
AH	0.22	0.42

## Data Availability

The data used in this study can be requested from the corresponding author: Prof. Vito Telesca at vito.telesca@unibas.it.

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
