# Peer review of "Effects of Meteo-Climatic Factors on Hospital Admissions for Cardiovascular Diseases in the City of Bari, Southern Italy"

_healthcare, 2023, doi:10.3390/healthcare11050690_

Round 1
Reviewer 1 Report
This study analyzing relationship between the weather and hospital admissions for cardiovascular disease, especially the delayed effect of temperature (shown in Figures 7 and 8), is interesting. However, it is rather difficult to follow the application of lag time model and calculation method related to the model. The authors should explain them in detail and with ease.
The authors analyzed cardiovascular disease as a whole. However, there are many diseases as shown in the manuscript. Their severity and urgency greatly vary. Specific patients developed ischemic cardiac events including cardiac infarction should be analyzed. These patients require immediate recanalization therapy in high-volume center for cardiovascular diseases. Once treatment works, the patients will fully return to their everyday life. Therefore, the information related to this type of diseases will provide great help for the daily medical practice, leading to valuable use of medical resources.
Author Response
Point 1: This study analyzing relationship between the weather and hospital admissions for cardiovascular disease, especially the delayed effect of temperature (shown in Figures 7 and 8), is interesting. However, it is rather difficult to follow the application of lag time model and calculation method related to the model. The authors should explain them in detail and with ease.
Response 1: Thank you for sharing your feedback on our analysis of the relationship between weather and hospital admissions for cardiovascular disease. We are pleased to inform you that we have prepared a revised version of the manuscript that incorporates your suggestions.In this new version of the manuscript, we have sought to provide a more detailed and accessible explanation of the application of the lag time model and the associated calculation method. We hope that this more detailed explanation will clarify the methodology used in our study and make it easier for readers to understand how we arrived at our conclusions.
Point 2: The authors analyzed cardiovascular disease as a whole. However, there are many diseases as shown in the manuscript. Their severity and urgency greatly vary. Specific patients developed ischemic cardiac events including cardiac infarction should be analyzed. These patients require immediate recanalization therapy in high-volume center for cardiovascular diseases. Once treatment works, the patients will fully return to their everyday life. Therefore, the information related to this type of diseases will provide great help for the daily medical practice, leading to valuable use of medical resources.
Response 2: The analyzed data concerns emergency room admissions; however, such data does not contain all the necessary information to analyze specific pathologies within the cardiovascular category.

Author Response
Point 1: Lines 187-191: The reviewer thought some other meteorological variables (namely, daily maximum atmospheric pressure, daily minimum atmospheric pressure, daily mean precipitation, daily mean water vapor pressure, daily mean total cloud amount and daily mean wind velocity) should be taken into consideration as well. Then the authors could select the climatic variables that can most influence CVD risks from them.
Response 1: Thank you for taking the time to review our work and for your suggestion to use additional meteorological variables. Unfortunately, the available data, in terms of both quantity and quality, does not permit the use of further variables. This limited availability of data is a limiting factor for our analysis and prevents us from exploring additional variables. We understand the importance of using a comprehensive set of variables, but in this case, it was not feasible due to the available data.
Point 2: Lines 262-264: The authors should use the ensemble empirical mode decomposition (EEMD) method (Huang et al. 1998; Huang and Wu 2008; Wu and Huang 2009) to verify the robustness of the results.
Response 2: Thank you for your assessment and your request to use the Ensemble Empirical Mode Decomposition (EEMD) method to verify the robustness of the results. Our goal was to identify a simple method to verify that signal decomposition would lead to a noticeable improvement in results and allow the use of ML techniques for simulating these processes. This was accomplished by verifying the validity of STL and also evaluating the stationarity of the data series using the Dickey-Fuller method. Our intention in a future work is to apply more complex decomposition methods, such as the Ensemble Empirical Mode Decomposition (EEMD), and this has been included in the future prospects of this work, along with references.
Point 3: Lines 398-401: The results derived from DLNM should be discussed and compared with concrete previous studies. Besides, the authors should explain why there are no significant cold effects
Response 3: Several studies have utilized the Distributed Lag Non-Linear Model (DLNM) approach to examine the association between PM2.5 exposure, temperature and health outcomes in China, in order to explore the correlation between air pollution and human health, demonstrating the efficacy of the DLNM method in these kinds of analyses.The study by Slama et al. (2020), conducted in five cities in Poland over a period of four years, explored the association between air pollution peaks and respiratory disease hospitalizations. The results revealed a positive impact of the DLNM method in this type of analysis. The particulate matter (PM2.5) was found to be the most significant pollutant. Additionally, Guo et al. (2014) found that both cold and hot temperatures increase the risk of mortality in various countries and regions, with the temperature associated with the lowest mortality being around the 75th percentile, ranging from 66th (Taiwan) to 80th (UK) percentiles. The DLNM method was also used to analyze the effects of temperature on respiratory disease hospitalizations, and the results were remarkable. These studies highlight the applicability of the DLNM method and reinforces its positive impact in similar analyses. Finally, we are pleased to inform you that we have prepared a revised version of the manuscript that incorporates your suggestions

Round 2
Reviewer 1 Report
The manuscript was revised well according to my comments. I have no further criticism.
Reviewer 2 Report
I suggest the current version should undergo MDPI's Language Editing Services before this manuscript can be published in Healthcare.